# The GATA-Type Transcriptional Factor Are1 Modulates the Expression of Extracellular Proteases and Cellulases in *Trichoderma reesei*

**DOI:** 10.3390/ijms20174100

**Published:** 2019-08-22

**Authors:** Yuanchao Qian, Yu Sun, Lixia Zhong, Ningning Sun, Yifan Sheng, Yinbo Qu, Yaohua Zhong

**Affiliations:** 1State Key Laboratory of Microbial Technology, Institute of Microbial Technology, Shandong University, Qingdao 266237, China; 2Shandong Institute for Food and Drug Control, Jinan 250101, China; 3College of Life Science, Northeast Agricultural University, Harbin 150030, China

**Keywords:** *Trichoderma reesei*, GATA factors, Are1, Cellulase, Protease

## Abstract

*Trichoderma reesei* is a biotechnologically important filamentous fungus with the remarkable ability to secrete large amounts of enzymes, whose production is strongly affected by both the carbon and nitrogen sources. While the carbon metabolism regulators are extensively studied, the regulation of enzyme production by the nitrogen metabolism regulators is still poorly understood. In this study, the GATA transcription factor Are1, which is an orthologue of the *Aspergillus* global nitrogen regulator AREA, was identified and characterized for its functions in regulation of both protease and cellulase production in *T. reesei*. Deletion of the *are1* gene abolished the capability to secrete proteases, and complementation of the *are1* gene rescued the ability to produce proteases. Quantitative RT-PCR analysis revealed that the transcripts of protease genes *apw1* and *apw2* were also significantly reduced in the Δ*are1* strain when grown in the medium with peptone as the nitrogen source. In addition, deletion of *are1* resulted in decreased cellulase production in the presence of (NH_4_)_2_SO_4_. Consistent with the reduction of cellulase production, the transcription levels of the major cellulase genes, including *cbh1*, *cbh2*, *egl1*, and *egl2,* were dramatically decreased in Δ*are1*. Sequence analysis showed that all promoter regions of the tested protease and cellulase genes contain the consensus GATA elements. However, the expression levels of the major cellulase transcription activator Xyr1 and the repressor Cre1 had no significant difference between Δ*are1* and the parental strain QM9414, indicating that the regulatory mechanism deserves further investigation. Taken together, these results demonstrate the important role of Are1 in the regulation of protease and cellulase production in *T. reesei*, although these processes depend on the kind of nitrogen sources. The findings in this study contribute to the understanding of the regulation network of carbon and nitrogen sources in filamentous fungi.

## 1. Introduction

Filamentous fungi have the excellent capacity to secrete a wide range of enzymes, such as cellulase and protease, involved in the degradation and recycling of complex biopolymers from organisms [1,2]. These hydrolytic enzymes play an important role in nutrition intake for fungi by releasing carbon and nitrogen locked in insoluble macromolecules [3]. Fungi have developed efficient and sensitive genetic regulatory systems, which were highly depended on transcription factors, to enable them to synthesize suitable enzymes for rapidly responding to the fluctuating carbon and nitrogen sources in the environment [4]. So far, many different types of transcription factors, which are involved in regulating the production of secreted enzymes, have been identified in fungi [5,6,7,8,9,10,11].

The filamentous fungus *Trichoderma reesei* has the capacity to secrete large amounts of cellulolytic enzymes for deconstruction of plant biomass substrates [12]. The expression of cellulase genes is tightly blocked by the carbon catabolite repression (CCR), which was mediated by the repressor Cre1, in the presence of the preferred carbon sources, such as glucose [5]. While, the transcripts of the cellulase genes were activated by the transcription activator Xyr1 in the presence of the non-preferred carbon sources, such as Avicel [9]. However, two recent findings have suggested that among the over 400 CAZy genes in *T. reesei*, only a few genes appear to be directly regulated by Xyr1 and Cre1 [13,14]. According to these reports, it can be inferred that additional regulatory factors that regulate the expression of cellulase genes should be found in the genome of *T. reesei*.

Like the CCR that is the important mechanism employed in the utilization of carbon, NMR (nitrogen metabolism repression) is a global regulatory mechanism that modulates the expression of catabolic genes in response to available nitrogen sources in the environment [15]. In filamentous fungi, the global response to limiting nitrogen conditions is mediated by positively acting GATA-type zinc finger proteins, of which AREA from *Aspergillus nidulans* and NIT2 from *Neurospora crassa* are the most extensively characterized [16,17]. Another zinc finger protein, AREB, possibly acts as a potential negative regulator of nitrogen catabolism under nitrogen-limiting conditions and may be involved in fungal growth, conidial germination, and asexual development [18]. Particularly, the roles of AREA homologues go beyond the regulation of the pathway involved in nitrogen utilization [19]. AREA homologues are required for the expression of genes involved in the biosynthesis of secondary metabolites, such as aflatoxin in A*spergillus flavus* [20], gibberellin in *Gibberella fujikuroi* [21], and deoxynivalenol in *Fusarium graminearum* [22,23]. AREA homologues are also involved in chromatin accessibility and are essential for the full virulence of some plant pathogenic fungi [24,25,26,27]. Furthermore, AREA homologues can activate the expression of protease genes in *Candida albicans* and probably participate in the regulation of cellulase expression in *A. nidulans* [28,29,30]. Given the importance of *T. reesei* as a major producer of commercial cellulase product and the absence of research published concerning the function of GATA-type zinc finger regulators in *T. reesei*, it is of great significance to investigate the roles of AREA/NIT2 orthologues in *T. reesei*.

Here, the GATA-type transcriptional activator Are1, which regulates the transcription of extracellular enzyme genes, was identified and characterized in *T. reesei.* Deletion and complementation studies of the *are1* gene demonstrated that the *are1* is absolutely necessary for activating the expression of extracellular proteases in *T. reesei* in the presence of the nonpreferred nitrogen sources, such as skim milk. Furthermore, the cellulolytic ability and transcriptional level analysis also revealed that Are1 is involved in the regulation of cellulase expression in the presence of the preferred nitrogen sources, such as ammonium. These findings help to elucidate the regulation network of carbon and nitrogen sources in filamentous fungi.

## 2. Results

### 2.1. Generation of the are1, are2, and are3 Gene-Deletion Strains in T. reesei

Inspection of the *T. reesei* genome database (available online: https://genome.jgi.doe.gov/Trire2/Trire2.home.html) with TBLASTX using the classic GATA-type transcription factor, *Aspergillus nidulans* AREA (GenBank accession no. CAA36731.1), as the query revealed three orthologues, which were named as Are1 (protein ID: tre76817), Are2 (protein ID: Tr_12017), and Are3 (protein ID: Tr_4231), respectively. The Are1-encoding gene *are1* contains a predicted 2835-bp open reading frame that is interrupted by two introns and encodes a protein of 944 amino acids, whereas Are2 and Are3 consist of putative 429 and 554 amino acids, respectively. The phylogenetic analysis of AREA/NIT2-related orthologues from different fungal species showed that the *T. reesei* Are1 is most close to NIT2 of *N. crassa*, while Are2 and Are3 are most similar to AREB of *A. nidulans* (Figure 1A). Further dissection of the amino acid sequences of these proteins demonstrated that they all contained the conserved Cys_2_-Cys_2_ zinc finger motif (C-X_2_-C-X_17_-C-X_2_-C) (Figure 1B and Appendix A), indicating the putative conserved role of these orthologues in filamentous fungi. In order to further confirm that the *T. reesei* Are1 is an orthologue of AreA/NIT2, another phylogenetic analysis was performed focusing on the orthologues of AreA/NIT2 (Appendix A). The *T. reesei* Are1, *F. fujikuroi* AreA, *M. grisea* NUT1, and *N. crassa* NIT2 form a well-supported clade, while the other AreA/NIT2 orthologues of filamentous fungi, including *A. nidulans* AreA, *A. niger* AreA, *A. oryzae* AreA, *A. parasiticus* AreA, *P. chrysogenum* NRE, and *P. roqueforti* Nmc fall within another supported clad. In addition, the transcription levels of *are1* under various nitrogen sources were investigated. It was found that there was no significant difference (*p* > 0.05) in the *are1* transcript formation between the preferred nitrogen substance (NH_4_)_2_SO_4_ and the non-preferred nitrogen source peptone or no nitrogen source (Appendix A), indicating that the transcription of *are1* was possibly nitrogen source-independent.

To investigate the function of Are1, Are2, and Are3 in *T.reesei*, the *are1*, *are2,* and *are3* deletion strains were generated by homologous recombination, respectively (Appendix A). A total of four putative *are1*-deletion mutants were constructed and verified by PCR analysis (Appendix A). Then, one of these candidates, named as Δ*are1*, was selected for a further Southern blot assay (Appendix A). The *Eco*RI/*Kpn*I-digested or *Eco*RI-digested genomic DNA was hybridized with the *are1* probe and showed a 2.7 kb or 5.2 kb fragment for the Δ*are1* strain, respectively, while a 4.0 kb or 6.5 kb fragment was shown for the parental strain *T. reesei* QM9414, respectively (Appendix A). These results suggested that the *are1* gene was successfully deleted in *T. reesei*. Similarly, both of the Δ*are2* and Δ*are3* strains were constructed and verified by PCR (Appendix A). Then, the *are1*, *are2*, and *are3* gene-deletion strains were used for further investigation.

### 2.2. Disruption of are1 Abolishes Protease Production in T. reesei

Previous studies showed that the production of extracellular proteases was tightly regulated by AREA/NIT2 in several fungi, such as *C. albicans,* and *P.marneffei* [27,28,29,31] and deletion of *areA* resulted in the disability to produce the extracellular proteases when the deletion strains were grown on the skim milk agar plates [31]. In order to identify whether Are1, Are2, or Are3 is involved in the regulation of the extracellular protease production in *T. reesei*, the parental strain QM9414 and the deletion strains (Δ*are1*, Δ*are2,* Δ*are3*) were cultured on the skim milk plates. As shown in Figure 2, there was a clear proteolytic halo around the colony of QM9414 with skim milk as the sole nitrogen source, while there was no proteolytic halo around the colony of QM9414 in the presence of (NH_4_)_2_SO_4_, suggesting that the extracellular protease production in *T. reesei* was repressed by preferred nitrogen sources, such as (NH_4_)_2_SO_4_. Disruption of *are2* or *are3* still generated the proteolytic halo around the colony on the medium using skim milk as the sole nitrogen source, whereas Δ*are1* did not grow on this medium, suggesting that Are1 may play a critical role for the uptake of nitrogen on this medium.

To determine if the re-introduction of Are1 to the *T. reesei* genome was sufficient to rescue the protease-defective phenotype in Δ*are1*, the *are1* complementation (*are1*-Com) and the P*gpdA*::*are1* fusion (*are1*-Fusion) strains were constructed and confirmed by PCR analysis (Appendix A). Then, the ability of the *are1*-Com and the *are1*-Fusion strains to produce extracellular proteases in the skim milk medium was tested. As shown in Figure 3, there was a clear proteolytic halo around the colony of the *are1*-Com strain with skim milk as the sole nitrogen source, indicating that complementation of *are1* with the *are1* gene could restore the production of proteases on the skim milk medium. In addition, it was found that the *are1*-Fusion strain exhibited a larger hydrolytic halo around the colony than that of QM9414, suggesting that expression of *are1* under the control of P*gpdA* could increase the production of proteases in *T. reesei*. Considering that Δ*are1* could not grow on the medium using skim milk as the sole nitrogen source, the Δ*are1* strain were cultured on the skim milk agar plate plus peptone as the complex nitrogen source to further determine whether the protease-defective phenotype was caused by the impaired growth. As shown in Figure 3, there was no proteolytic halo formed around the colony of Δ*are1* compared to that of the parental strain QM9414, indicating that the disability of Δ*are1* to produce proteases was not due to the defective growth, but to the absence of Are1. Taken together, these results demonstrated that Are1 has an essential function in protease production in *T. reesei.*

### 2.3. Are1 is a Key Factor in Modulation of Protease Expression in T. reesei

It has been shown that the aspartic proteases APW1 and APW2 may be the major proteases produced by *T. reesei* when exposed to the non-preferred nitrogen sources, such as BSA [32]. Here, the transcription level of *apw1* in the *T. reesei* QM9414 cultured with peptone as the sole nitrogen source was increased by a fold-change of 156 with a *p*-value = 1.17 × 10^−6^ when compared to that with peptone plus (NH_4_)_2_SO_4_ (Figure 4A). In the same conditions, *apw2* showed an increase by a fold-change of 5.9 with a *p*-value = 1.10 × 10^−5^ (Figure 4B), indicating that the expression of proteases in *T. reesei* was repressed by ammonium, which was consistent with the results obtained from the skim milk plates. When *are1* was deleted, it was found that the transcription levels of *apw1* and *apw2* were decreased sharply in the Δ*are1* strain compared to that in QM9414 (a 99.87% decrease with a *p*-value = 1.16 × 10^−6^ for *apw1* and an 83.51% decrease with a *p*-value = 1.16 × 10^−6^ for *apw2*) using peptone as the sole nitrogen source, demonstrating that Are1 positively regulated the expression of proteases in the non-preferred nitrogen sources, such as peptone (Figure 4). Moreover, the transcription of *apw1* was up-regulated in the Δ*are1* strain when ammonium was added (Figure 4A), suggesting that deletion of *are1* could relieve the repression of protease expression by ammonium. It was known that Are1 orthologues recognize the DNA motif, 5′-HGATAR-3′, located in the promoters of AREA-regulated genes [33]. As shown in Figure 4C, the promoter sequences of the protease-genes, *apw1* and *apw2,* contain the conserved motif, indicating that Are1 may participate in the regulation of protease gene expression through directly binding to the protease gene promoters.

### 2.4. Cellulase Production is Reduced in the Δare1 Strain

The filamentous fungus *T. reesei* has the ability to secrete large amounts of cellulolytic enzymes and thus is the main industrial source for cellulase production [12]. So, the effect of deletion of the *are1* gene on cellulase production was evaluated. Firstly, the Δ*are1* strain was grown on the CMC plates containing sodium carboxymethyl cellulose (CMC-Na) as the sole carbon source to test its cellulolytic ability. It was found that the cellulolytic halo around the colony of Δ*are1* was much smaller than that of the parental strain QM9414, while introduction of a copy of *are1* or expression of *are1* under the control of P*gpdA* both restored the cellulolytic defect of Δ*are1* (Figure 5A), suggesting that Are1 may regulate the production of cellulase in *T. reesei.* To further examine the influence of *are1* deletion on cellulase production, the Δ*are1*, *are1*-Com, and *are1*- Fusion strains were cultured in cellulase-inducing medium (CM) at 30 for 7 days. Then, the fermentation supernatants were collected at a specified time interval and the activities of total cellulase (FPA) and endoglucanase (EG) were measured. As shown in Figure 5B,C, both FPA and EG activities were significantly decreased in Δ*are1* compared to that of the parental strain QM9414. Meanwhile, complementation and expression of *are1* under the control of P*gpdA* completely restored the ability to produce cellulases. Therefore, these results suggested that Are1 also plays a role in the production of cellulases in *T. reesei.*

### 2.5. Are1 is Involved in Regulation of the Expression of Cellulases in T. reesei

The fact that cellulase production was greatly decreased in the Δ*are1* strain made us to wonder whether *are1* deletion leads to the down-regulated expression of the cellulase genes. Then, the transcription levels of the major cellulase genes *cbh1, cbh2, egl1*, and *egl2* were investigated by RT-qPCR (Figure 6). It was discovered that the transcripts of *cbh1, cbh2, egl1*, and *egl2* were significantly reduced in the Δ*are1* strain compared with the parental strain QM9414 in the presence of ammonium. However, there was no significant difference (with *p* > 0.05) in the transcription levels of cellulase genes between Δ*are1* and QM9414 when cultured in the medium containing the peptone as the sole nitrogen source (Figure 6). These results suggested that *are1* possibly positively regulates the expression of cellulase genes in the preferred nitrogen source, such as ammonium.

Considering peptone is an organic compound that could be used as both nitrogen and carbon sources and may somehow affect cellulase expression [34], then (NH_4_)_2_SO_4_ was chosen as the sole nitrogen source for cultivation of *T. reesei* to examine the effect of *are1* deletion on cellulase-gene expression. As shown in Figure 7A, the deletion of *are1* significantly decreased the expressions of *cbh1* and *cbh2* in Δ*are1.* This result further demonstrated that Are1 may sense the preferred nitrogen source, such as ammonium, and act as a regulator to modulate cellulase gene expression. It was known that the expression of cellulase genes was positively regulated by the major cellulase regulator, Xyr1, and negatively regulated by the carbon catabolite repressor, Cre1, in *T. reesei* [5,9]. Here, to investigate whether Are1 modulates cellulase-gene transcription through these regulators, the transcript abundances of *xyr1* and *cre1* in the Δ*are1* strain were evaluated. As shown in Figure 7B, there were no remarkable differences (with *p* > 0.05) in the transcriptional levels of *xyr1* and *cre1* between Δ*are1* and the parental strain QM9414. Moreover, it was found that the promoter sequences of *cbh1* and *cbh2* also contained the Are1-binding motif, 5′-HGATAR-3′ (Figure 7C), indicating that Are1 may act as a regulator through binding to the cellulase gene promoters. Taken together, these results demonstrated that Are1 is also involved in regulation of the expression of cellulases in *T. reesei*.

## 3. Discussion

Nitrogen metabolite repression mediated by Are1 homologues is widely distributed among filamentous fungi [26,35]. In this study, we identified the nitrogen regulatory factor Are1 from *T. reesei* and investigated its function in regulation of both protease and cellulase production. The predicted protein Are1 shares extensive regions of homology with the known positive nitrogen regulatory proteins AREA of *A. nidulans* and NIT2 of *N. crassa*, which are highly conserved fungal GATA-type zinc finger proteins. Δ*are1* was unable to grow on medium containing skim milk as the sole nitrogen source, which was similar to the AreA mutant of *A. nidulans* and the NIT2 mutant of *N. crassa* [16,17], suggesting that Are1 is required for utilization of the non-preferred nitrogen in *T. reesei.* However, unlike the AreA and NIT2 mutants, the Δ*are1* strain showed a severely impaired growth on the medium with ammonium, indicating the different role of *are1* on nitrogen utilization from AreA in *A. nidulans* and NIT2 in *N. crassa*. It is possible that deletion of *are1* could affect the uptake of ammonium in *T. reesei*. In addition, Δ*are1* displayed a similar growth to that of the parental strain on the agar plates containing peptone as the nitrogen source. Considering peptone contains free amino acids, it is likely that *are1* deletion had no effects on the transportation or metabolism of some amino acids in *T. reesei.* This is consistent with the results of Bugeja et al. (2012), who reported that AreA is not required for the utilization of some amino acids, such as glutamate and alanine, in *P. marneffei* [31].

Are1 orthologues have been shown to be transcription activators of the extracellular proteases in some filamentous fungi [26,28,29,31] and it is therefore not surprising that the ability to produce proteases on the skim milk agar plates was abolished in the *T. reesei* strain Δ*are1*, confirming that Are1 was a key factor to secrete proteases in *T. reesei*. In addition, the expression of the aspartic protease genes, such as *apw1*, was induced by peptone and repressed by ammonium in *T. reesei,* which is consistent with the results reported for other fungi where the expression of proteases was induced by non-preferred nitrogen sources and repressed by the favored nitrogen sources [31,35]. Moreover, the transcription level of *apw1* was increased in the presence of ammonium in Δ*are1*, indicating that mutation of Are1 resulted in derepression of protease gene expression, which is similar to that in the AreA mutant of *A. nidulans* [36].

Evidence is emerging that the cellulase production was reduced in the strain containing the loss function of *areA* in *A. nidulans* [30]. In the present study, it was found that the transcripts of the cellulase genes in Δ*are1* were significantly reduced compared to that in the parental strain QM9414 in the presence of ammonium. However, there was no significant difference in the cellulase-gene transcripts between Δ*are1* and QM9414 when cultured in the medium containing peptone as the sole nitrogen source. These results suggested that the regulation of cellulase gene expression by Are1 depends on the kind of the nitrogen sources in the environment for *T. reesei*. Meanwhile, the conserved Are1-binding motif was also found in the promoters of the cellulase genes, *cbh1* and *cbh2*. Although the binding function of Are1 needs to be confirmed experimentally, such as EMSA, the decreased cellulase activities in Δ*are1* indicated that Are1 is involved in regulation of cellulase expression in *T. reesei*. The transcript abundances of the major cellulase regulators, Xyr1 and Cre1, in Δ*are1* were comparable to that of the parental strain QM9414, suggesting that Are1 may act directly on the cellulase gene promoter to regulate the cellulase production, independent of Xyr1 and Cre1.

Carbon catabolite repression (CCR) and nitrogen metabolite repression (NMR), which cooperatively ensure glucose and ammonium are utilized preferentially by preventing the expression of genes required for the metabolism of less preferred carbon and nitrogen sources, are the most important nutrient control laws in the microbial world [37,38]. Protease, as an important participant in the nitrogen source cycle, has been proved to be regulated by nitrogen regulators Are1/AreA/NIT2. Similarly, cellulase, as an important decomposer of lignocellulose (complex carbon source), is also strictly depended on the status of carbon sources. In this study, the expression of both cellulases and proteases for the first time was found to be coupled under the control of the GATA-type transcriptional factor Are1. However, the regulatory mechanisms between the cellulase and protease production were significantly different. When the nitrogen source was sufficient, Are1 could activate the expression of cellulases and inhibit the expression of proteases. When the nitrogen source is insufficient, the expression of proteases is activated and expression of cellulases is inhibited. Taken together, these results demonstrated that Are1 probably acts as a mediator to regulate the expression of cellulases and proteases to facilitate balancing the utilization of carbon and nitrogen sources.

## 4. Materials and Methods

### 4.1. Strains and Culture Conditions

The filamentous fungus *T. reesei* QM9414 (ATCC 26921) was used as the parental strain for construction of all the deletion strains in this study. Relative strains were grown on potato dextrose agar (PDA) plates (200 g/L potato, 20 g/L glucose, 20 g/L agar) at 30 °C for 5–7 days to harvest spores. Then, the spores were collected and then 10^8^ of spores were pre-cultured in 150 mL of minimal medium or induction medium (CPM, Minimal uridine medium with 2% Lactose or Avicel) at 30 °C for 36 h and, subsequently, 1g of mycelia was transferred into 150 mL of CPM for cellulase production [39]. The plasmid T-hph was used as the template to amplify the hygromycin B-resistant gene *hph*. The pyrithiamine-resistant gene *ptrA* was amplified from the plasmid T-ptrA. Minimal medium supplemented with 0.3 μg/mL pyrithiamine (Sigma, St. Louis, Missouri, USA) or 300 μg/mL hygromycin B was applied as a selective medium for screening the fungal transformants. The skim-milk agar plates were used to investigate the capacity of *T. reesei* strains to secrete proteases. The composition of skim-milk medium was MM supplemented with 2% skim milk plus different carbon sources or different nitrogen sources. The CMC plates were used to screen the strains showing cellulase (EG) activity. The CMC medium composition was as follows (g/L): 10 CMC–Na, 1 yeast extract, 5 triton X-100, and 20 agar.

### 4.2. Molecular Techniques

Primes were designed by using the primer premier 5.00 software (PREMIER Biosoft, Palo Alto, CA, USA). DNA fragments were purified using a Gel Extraction Kit (Omega, Norcross, GA, USA). Oligonucleotides synthesis and DNA sequencing were performed at Sangon Inc. (Shanghai, China). Oligonucleotides used in this study are listed in Appendix A. Multiple alignments of protein sequences were performed with ClustalW2 (http://www.ebi.ac.uk/Tools/msa/ clustalw2/). Phylogenetic analysis was inferred using the neighbor joining method and the software MEGA4.0. The *T. reesei* genome database ver2.0 (http://genome.jgi-psf.org/Trire2/Trire2.home.html) was used for DNA and protein predictions in this study.

### 4.3. Construction of are1, are2, and are3 Genes Deletion Strains

The disruption cassettes for gene disruptions were constructed by the double-joint PCR method previously described [40]. All primers used in this study were listed in Table 1. To construct the Are1 deletion strain, firstly, 1.6 kb- and 1.3 kb- DNA fragments corresponding to the 5′ and 3′ regions of the *are1* gene were amplified from the genome of *T. reesei* using the primer pairs Are1-1618-UF1/Are1-52-UR1 and Are1-232-DF1/Are1-1729-DR1, respectively (Table 1). The hygromycin B resistant gene *hph* (1.9 kb) was amplified from T-hph with primer pair Hph-F1/Hph-R1 (Table 1). Then, these PCR products were mixed and used as the template to amplify the deletion cassette by PCR using the nest primer pair Are1-1522-UF2/Are1-1426-UR2. A final product (Δ*are1::hph*, 4.4 kb) was obtained and then was transformed into protoplasts of *T. reesei* using the previously described method (Ghose, 1987). The *are1* deletion transformants were identified by PCR using the primer pairs Are1-1618-UF1/Y-hph-168-UR1 and Y-hph-121-DF1/Are1-1729-DR1. Similarly, another deletion cassette, Δ*are2*::prtA (5.4 kb) consisted of three fragments, the 5′ flanking region of *are2* gene (1.9 kb), the 3′ flanking region of *are2* gene (1.2 kb), and the *ptrA* gene (2.1 kb). These fragments were generated by the PCR method using the primer pairs Are2-2073-UF1/Are2-173-UR1, Are2-70-DF1/Are2-1537-DR1 and ptrA-F1/ptrA-R1, respectively. Then these PCR products were mixed and were used as the template to amplify the *are2*-deletion cassette using the Are2-1732-UF2/Are2-1403-UR2 as the primer pair. Δ*are2* mutants were identified by PCR using primer pairs Are2-2073-UF1/Yan-ptrA-UR1 and Yan-ptrA-DF1/Are2-1537-DR1, which yielded the 2.4 kb- and 1.5 kb- fragments, while there was no fragment for the parental strain. The Δ*are3*::ptrA cassette for *are3* deletion was constructed by the same method. Then, the cassette was transformed into the protoplasts of *T. reesei*. The Δ*are3* transformants were analyzed by PCR using the primer pairs Are3-2073-UF1/Yan-ptrA-UR1 and Yan-ptrA-DF1/Are1-1949-DR1, which produced the 2.1 kb- and 1.6 kb- fragments in the transformants.

### 4.4. Complementation of are1 in the Δare1 Strain

To construct the complementation strain, a PCR-amplified product (5.0 kb) containing the entire Are1 encoding gene was amplified by primer pair Are1-1064-UF/Are1-535-DR. To constitutively express the *are1* gene, the *are1* encoding region (4.1 kb) and 3′flanking region (1.5 kb) were amplified by PCR with primer pairs Are1-1-F/Are1-923 DR1 and Are1-1618-UF1/Are1-52-UR1 and fused with the constitutive *A. nidulans gpdA* promoter (1.0 kb). The promoter was obtained from the plasmid pAB4-1 with primer pair gpdA-S/gpdA-A (Table 1). Then, these DNA fragments were fused by double-joint PCR with the nested primer pair Are1-1522-UF2/Are1-340-DR2 (Table 1), resulting in a 6.2 kb product. The complementation cassette and constitutive expression cassette were transformed into the protoplasts of Δ*are1* strain. Candidates were selected on regeneration medium containing 2 g/L NaNO_3_. Transformants with the successful complementation and constitutive expression of *are1* were confirmed by PCR using primer pair ARE1-1-F/ARE1-1072-R, which expected a 1.0 kb product in the mutant.

### 4.5. RNA Extraction and RT-qPCR Analysis

For RNA extraction, 10^8^ spores were inoculated in minimal medium with 1% glucose and cultured for 36 h at 30 °C. Then, the mycelia were harvested and transferred into the induction medium (with peptone) supplemented with or without (NH_4_)_2_SO_4_. Considering the peptone used as nitrogen and carbon sources, the mycelia also transferred into the induction medium (without peptone) containing the (NH_4_)_2_SO_4_ as the sole nitrogen source. Then, the mycelia were harvested after 10 h of cultivation and then total RNA were isolated using the RNAiso™ reagent. The integrity of the extracted RNA was evaluated by agarose gel electrophoresis and then the total RNA was treated with DNAaseI. cDNA was synthesized from total RNA using a PrimeScript RT reagent kit, following the manufacturer’s description. The transcript levels of the *apw1*, *apw2*, *cbh1*, *cbh2*, *egl1*, and *egl2* were detected using an RT-PCR Kit with actin gene as a control. All of these primers used for PCR are listed in Table 1. The amplification efficiency of each gene was determined, which were between 93% and 102%. RT-qPCR analysis was performed on a LightCycler 480 System (Roche Diagnostics, Germany). The procedure of RT-qPCR was as follows: Initial denaturation of 1 min at 95 °C, followed by 40 cycles of 5 s at 95 °C and 20 s at 60 °C. Melting curve analysis from 65 to 95 °C was performed to confirm the specifics of the applications. Light-Cycler480 software 1.5.0 was used for the Ct value calculation. Data analysis was performed using the relative quantitation/comparative CT (ΔΔCT) method and were normalized to an endogenous control (actin). Three biological replicates were performed for each analysis and the results are shown as the mean and SD from the replicates. Statistical analysis was performed using the Student’s *t*-test analysis.

### 4.6. Detection of Cellulase Activity

The filter paper activity (FPA) was measured using Whatman No. 1 filter paper as a substrate, as described by Ghose (Ghose, 1987) [41]. The reaction mixtures containing 50 mg of Whatman No. 1 filter paper, 1.5 mL of 50 mM citrate buffer (pH 4.8), and 500 μL of the suitably diluted enzyme fractions were incubated at 50 °C for 60 min. EG activity was assayed with CMC–Na as a substrate. The enzyme reactions were performed in 2 mL of 1% CMC–Na (pH 4.8) at 50 °C for 30 min. The amount of reducing sugar released was determined using the DNS method. One unit of enzyme activity was defined as the amount of enzyme releasing 1 μmol of glucose per minute. Three biological triplicates were designed in all experiments.

## 5. Conclusions

In this study, the GATA-type transcription factor Are1 in *T. reesei* was identified and characterized for its function in the regulation of both protease and cellulase production. The *T. reesei* Δ*are1* was unable to produce extracellular proteases on the skim milk agar plates and showed a significant decrease in cellulase activity on the CMC agar plate. Further transcriptional analysis revealed that the regulation of protease and cellulase gene expression by Are1 highly depended on the kind of nitrogen sources. Sequence analysis showed that all promoter regions of the tested protease and cellulase genes contain the Are1-binding motif. These findings suggested that Are1 plays an important role both in the regulation of protease and cellulase production and may function in the manner of acting directly on the promoters of the target genes. To our knowledge, this is the first report of the function of Are1 in *T. reesei,* which provides new insights into the role of the GATA-type transcription factors in fungi.

## Figures and Tables

**Figure 1 ijms-20-04100-f001:**
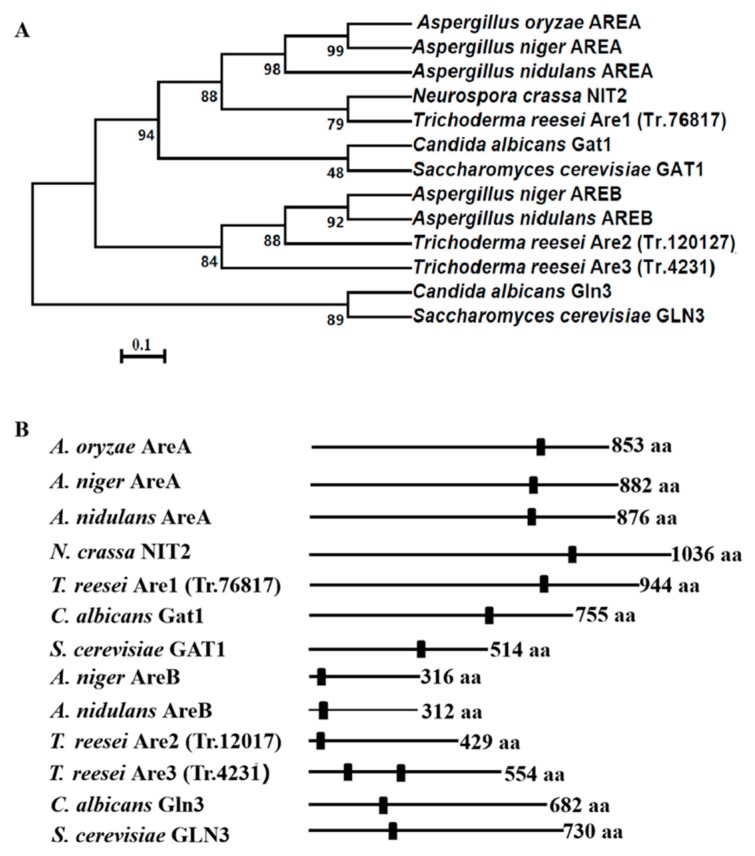
Homologous analysis of the GATA regulators Are1 (Trire2.76817)*,* Are2 (Trire2.120127), and Are3 (Trire2.4231) in *T. reesei*. (**A**) Phylogenetic analysis applying the neighbor joining method, including Are1 orthologue sequences from distinct fungi. GenBank accession numbers for the orthologue proteins are as follows: *C. albicans* (Gat1, AAP50501.1 and Gln3, AORE29836.1); *N. crassa* (NIT2, P19212.2); *A. oryzae* (AREA, AAK08066.1); *A. niger* (AREA, CAA68196.1 and AREB, XP_001399365.2); *A. nidulans* (AREA, CAA36731.1 and AREB, AAG49353), and *S. cerevisiae* (GAT1, KZV11580.1 and GLN3, KZV11792.1). The bar marker indicates the genetic distance, which is proportional to the number of amino acid substitutions. (**B**) Domain architectures in the Are1 orthologues from different fungi. The GATA zinc-finger domain (Pfam: PF00320) is represented by black squares. The length of each protein sequence (in amino acids) is indicated in the right.

**Figure 2 ijms-20-04100-f002:**
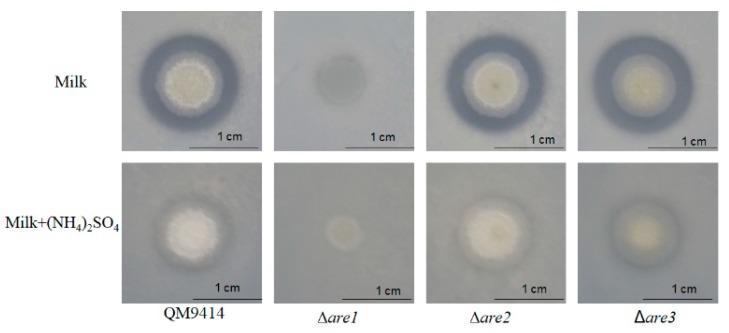
Growth of the *are1*, *are2*, and *are3* deletion strains on the plates containing 1% skim milk with or without 0.5% (NH_4_)_2_SO_4_ as the nitrogen source. The clear halos surrounding the colonies are due to extracellular protease activity.

**Figure 3 ijms-20-04100-f003:**
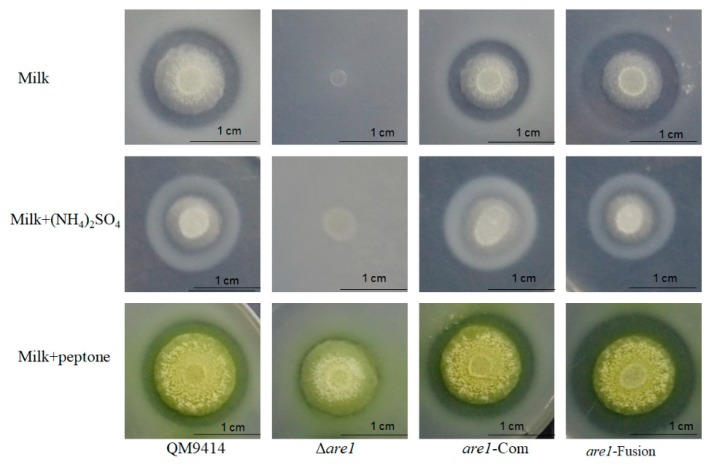
The extracellular protease production by the *are1* deletion strain Δ*are1*, the *are1* complementation strain *are1-*Com, the P*gpdA*::*are1* fusion stain (*are1*-Fusion), and the parental strain QM9414. The strains were grown on the plates containing skim milk, skim milk plus (NH_4_)_2_SO_4_, or skim milk plus peptone as the nitrogen source.

**Figure 4 ijms-20-04100-f004:**
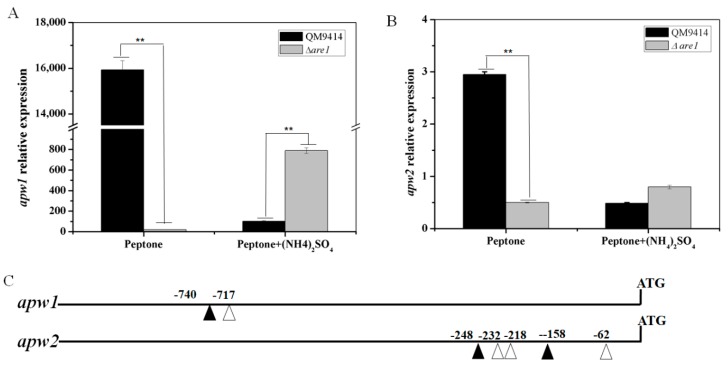
The transcription levels and the putative *cis*-regulatory elements of the extracellular protease-encoding genes *apw1* and *apw2* in *T. reesei*. (**A**) and (**B**) RT-qPCR analysis of the transcript abundances of *apw1* and *apw2* in the *T. reesei* strains Δ*are1* and QM9414, respectively. The measured quantity of the qPCR results in each of the samples in (**A**) and (**B**) was normalized with the actin gene. Error bars indicate the standard deviation and ** shows a *p*-value < 0.01. (**C**) Putative Are1-binding motifs in the promoter regions (within 1000 bp) of *apw1* and *apw2*. White triangle, HGATAR (where H stands for A, T or C, and R stands for A or G); Black triangle, YTATCD (where Y stands for T, G, or A, and D stands for T or C).

**Figure 5 ijms-20-04100-f005:**
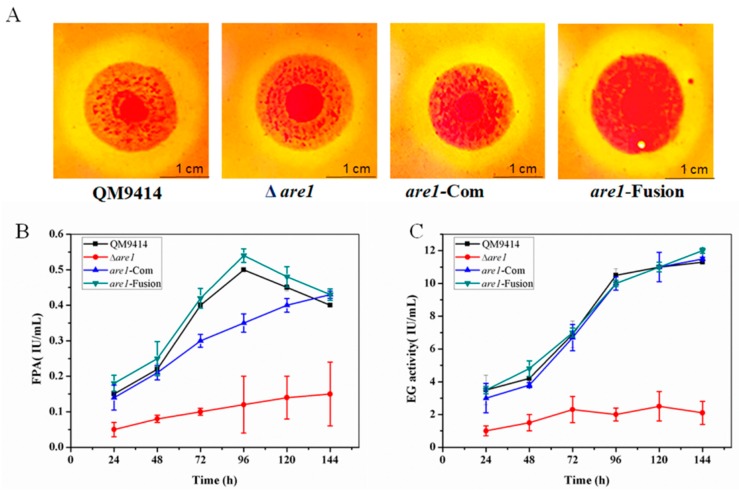
The cellulase production by the *are1* deletion strain Δ*are1*, the *are1* complementation strain *are1-*Com, the P*gpdA*::*are1* fusion stain *are1*-Fusion, and the parental strain QM9414. (**A**) Detection of the cellulolytic ability of the *T. reesei* strains on the carboxymethyl cellulose (CMC) agar plates. (**B**) and (**C**) The FPA (the activities of total cellulase) activity and the EG (endoglucanase) activity of the *T. reesei* strains during fermentation for cellulase production, respectively. All the strains were cultured in the cellulase production medium (CPM) for 144 h.

**Figure 6 ijms-20-04100-f006:**
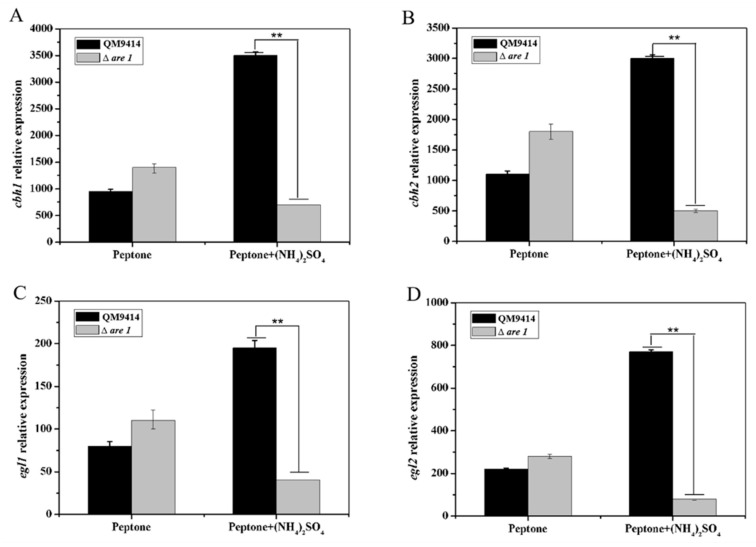
The transcription levels of the major cellulase genes *cbh1* (**A**), *cbh2* (**B**), *egl1* (**C**), and *egl2* (**D**) in the *T. reesei* strains Δ*are1* and QM9414. The strains were grown in CPM and RT-qPCR analysis was performed to detect the transcript abundance. Error bars indicate the standard deviation and ** shows a *p*-value < 0.01.

**Figure 7 ijms-20-04100-f007:**
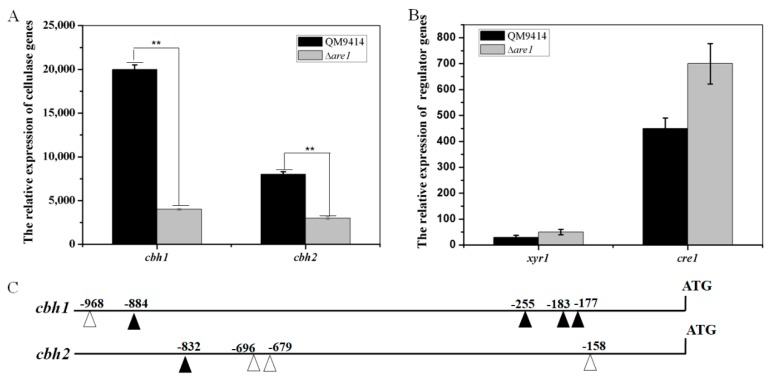
The possible regulation manner of the cellulase gene transcription by Are1 in *T. reesei*. (**A**) Validation of the transcription levels of the cellulase genes, *cbh1* and *cbh2*, in the *T. reesei* strains, which were grown in the MM medium using 10 mM (NH_4_)_2_SO_4_ as the sole nitrogen source. (**B**) The transcription levels of the major cellulase transcription factors, Xyr1 and Cre1, in the *T. reesei* strains. RT-qPCR analysis was performed to detect the transcript abundance in (**A**) and (**B**). Error bars indicate the standard deviation and ** shows a *p*-value < 0.01. (**C**) Putative DNA binding sites of Are1 in promoter regions (within 1000 bp) of *cbh1* and *cbh2*. White triangle, HGATAR (where H stands for A, T, or C and R stands for A or G); Black triangle, YTATCD (where Y stands for T, G, or A, and D stands for T or C).

**Table 1 ijms-20-04100-t001:** Oligonucleotides used in this study.

Primers	Sequences (5′-3′)	Employment
ARE1-1618-UF1	AAGCACTGGTTGTTGGTTGG	mutant construction
ARE1-52-UR1	TGCTCCTTCAATATCAGTTAAGGTCGAAGAAGGCTAATGGGGGAGAA	mutant construction
ARE1-232-DF1	AAATTCCGTCACCAGCCCTGGGTTGTGATACCTGGGTCTTTTGTGTG	mutant construction
ARE1-1729-DR1	TCTGGTTCCGATAGCCGA	mutant construction
ARE1-1522-UF2	CCCCTGTCCGTTAGCAGTTCAT	mutant construction
ARE1-1426-UR2	AGCATTTGGCATTTGCGAGAGA	mutant construction
Hph-F1	CGACGTTAACTGATATTGAA	mutant construction
Hph-R1	CAACCCAGGGCTGGTGACGG	mutant construction
ARE2 -2073-UF1	AGGGGCACTGGCAAATAT	mutant construction
ARE2 -173-UR1	GAAGCATAAAGTGTAAAGCCTGGGGGCCCGTTTTAGGCTTAGAG	mutant construction
ARE2-70-DF1	ATACAAACAAAGATGCAAGAGCGGCGTGGCAACGATGGATGT	mutant construction
ARE2-1537-DR1	ACCCTGAATGGTGTCCCTC	mutant construction
PtrA-F1	CCCCAGGCTTTACACTTTAT	mutant construction
ptrA-R1	CCGCTCTTGCATCTTTGTT	mutant construction
ARE3-1676-UF1	AGGGAGGCTGTCGGAGTG	mutant construction
ARE3-36-UR1	GAAGCATAAAGTGTAAAGCCTGGGGGGGCTTTTGTTGATGTTTCTC	mutant construction
ARE3-457-DF1	ATACAAACAAAGATGCAAGAGCGGCCCTGTCTATTCCTGAGTGT	mutant construction
ARE3-1949DR1	GGTATGAGTCGCAGGGAG	mutant construction
ARE3-1585-UF2	CGAACGAAGGTGGTGGAGAT	mutant construction
ARE3-1415-DR2	CCGCTTGTATGCTGGGGACT	mutant construction
Y-hph-121-DF1	ACTGAGGAATCCGCTCTTGG	mutant construction
Y-hph-168-UR1	ACTGCTTACAAGTGGGCTGA	mutant construction
Y-ptrA-UR1	CATTGTCGGTTGGTTTGG	mutant construction
Y-ptrA-DF1	ATGTAACGGTGGGGCATT	mutant construction
ARE1-1064-UF	TTATCGGTCAGTCTCCATCTTCA	mutant construction
ARE1-535-DR	TCCATTTTCCCAACATACTCCA	mutant construction
PgpdA-S	AGACCTAATACAGCCCCTAC	mutant construction
PgpdA-A	AACAGCTCCTCGCCCTTGCTCACCATATGTCTGCTCAAGCGGGGTAG	mutant construction
ARE1-3-F	GGCATGGACGAGCTGTACAAGGCAGCTGTCGGACCGCTT	mutant construction
ARE1-923-DR1	ATGAGCCCAAGAAAGACTGAAGAG	mutant construction
ARE1-340-DR2	CAGGCAGAGCAGAAAGAAATAAAAG	mutant construction
ARE1-1072-R	GTAGGGGCTTCTGTTCCA	mutant construction
ARE1-384DF	GAAGAACAGCCTCGGTGCG	probe
ARE1-1206DR	CTGCCTATCCCCAAGCGTC	probe
real-APW1-F1	AGGCACGGACAGAACGGCAGCT	RT-qPCR for *apw1*
real-APW1-R1	CGTTGGCGTAGTAGGCATCG	RT-qPCR for *apw1*
real-APW2-F1	GCGATGTCTACCACGATATTGTCTC	RT-qPCR for *apw2*
real-APW2-R1	TCAAGGCTGCCGACGATGTT	RT-qPCR for *apw2*
real-cbh1-F1	CCGAGCTTGGTAGTTACTCTG	RT-qPCR for *cbh1*
real-cbh1-R1	GGTAGCCTTCTTGAACTGAGT	RT-qPCR for cbh*1*
real-cbh2-F1	CTGGTCCAACGCCTTCTTCA	RT-qPCR for *cbh2*
real-cbh2-R1	GACCCAGACAAACGAATCCAG	RT-qPCR for *cbh2*
real-egl1-F1	CTCAGATGGACGAGAACGGG	RT-qPCR for *egl1*
real-egl1-R1	CTGGTGGCTAGTGTTGAGGG	RT-qPCR for *egl1*
real-egl2-F1	AACAAGTCCGTGGCTCCATT	RT-qPCR for *egl2*
real-egl2-R1	TCCGCTCCAACCAATACCTC	RT-qPCR for *egl2*
real-actin-F1	CCCAAGTCCAACCGTGAGA	RT-qPCR for *actin*
real-actin-R1	CAATGGCGTGAGGAAGAGC	RT-qPCR for *actin*

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
