# Peer review of "The GATA-Type Transcriptional Factor Are1 Modulates the Expression of Extracellular Proteases and Cellulases in Trichoderma reesei"

_ijms, 2019, doi:10.3390/ijms20174100_

Round 1

Reviewer 1 Report

In general, the article is well delineated, with interesting results, but needs some improvement.

line 75 – protease or proteases?

line 77 – regulation of cellulase expression, do you mean?

line 120 – protease or proteases?

Line 124 - protease or proteases?

Line 128 – I do believe that you should use the plural form when talking about protease production, since it is not only one protease that is synthesized by Trichoderma in such conditions. Please check this throughout the text.

Line 134 – separate inDeltaAre1

Line 145 – separate 3There

Line 148 – again, protease or proteases?

Figure 3 – Why did not you perform enzymatic assays for proteases using the different strains?

Line 156 – Why did you choose the apw1 and apw2 genes? The apw2 gene does not appear to be highly transcribed either in the presence of peptone or in peptone+ ammonium.

Line 165 - There is no statistical difference between the expression values of apw2 in the presence of peptone and peptone + ammonium. Therefore, it is not possible to affirm that Are1 is involved in the repression of the apw2 gene in the presence of ammonium.

Figure 4 – Start codon means first codon to be translated or first nucleotide to be transcribed?

Lines 207/208 – The transcripts of cbh1, cbh2, egl1 and egl2 were reduced in QM9414 or in deltaAre1?

Line 210 – correct the word cellulase and separate betweenΔare1

Line 219 – do you mean affect (effect)?

Line 220 - correct the word cellulase

Figure 7B – y axis. Correct the legend. Xyr1 and Cre1 are not cellulase genes.

Line 224 – cellulase genes were, not cellulase gene was…

Line 228 – please separate betweenΔare1

Line 269 – rewrite. Suggestion: a similar result was observed where the expression of the aspartic protease genes, apw1 and apw2, were induced by peptone…

Discussion section – Needs some improvements. There was a repetition in the description of the results, with little increase for the discussion regarding the literature. The figures should no longer be cited in the text in the discussion. Please, discuss better your results.

Reviewer 2 Report

Qian et al investigate the role of AreA/Nit2  orthologues in  Trichoderma reesei, and notably their role in cellulose metabolism. This point is of particular interest because this fungus is a major producer of hydrolytic enzymes for the industrial degradation of cellulose. The authors present data suggesting that Are1, the T. reesei orthologue of AreA/Nit2, is involved in cellulose metabolism, but most of their results need further work to provide reliable conclusions. Required modifications and experiments are presented below :

1.       The introduction needs some corrections and additions. Notably, AreB is mentionned in the results but is not presented in the introduction. Add a section about AreB with references. Other corrections are indicated below :

a.       Line 47 : references 5 to 11 are about T. reesei and its teleomorph Hypocrea jecorina. Therefore the generic designation « in fungi » indicating the focus of these references must be changed to «in T. reesei and its teleomorph Hypocrea jecorina ».

b.       Line 65 : bikaverin is not mentionned in reference 20. Cite another reference for regulation of bikaverin synthesis by AreA orthologue.

c.       Line 67 : add references for papers describing Candida homologues of AreA.

d.       Line 69. I have not found any mention of protease gene expression control nor of cellulase gene expression control in ref 26. This reference should be deleted here.

2.       Section 2.1, line 81 – 86. Search with TBLASTX reveals similarities with a query, not orthologues. The phylogenetic tree(Figure 1A) with so few  branches is not an argument for homology neither. Confirm that Are1 is a orthologue of AreA by additional analyses (reciprocal BLAST), or better, an analysis with FUNGIpath (http://fungipath.i2bc.paris-saclay.fr/)  [1]. Are2 and Are3 should be analyzed as well.

3.       Section 2.1, line 105-114. The authors have generated deletion strains for are1, are2 and are3. Confirmation of the deletion relies on southern blot, which is provided only for Δare1. The authors have demonstrated by PCR that crossing overs took place upstream and downstream of the are2 and are3 CDSs, but that is not sufficient to demonstrate that the genes are deleted.  The Δare2 and Δare3 strains are not affected for protease synthesis, however any conclusion about the role of these two genes is impossible since no data prove unambiguously that are2 and are3 genes are really deleted.

4.       Section 2.2, line 127-128. Further reading of the ms reveals that the Δare1 strain does not grow on medium with skim milk as the sole nitrogen source (see line 142 and 249). It is therefore not possible to see whether it produces a halo on this medium. I suggest to change this sentence to : « …, whereas Δare1 did not grow on this medium, suggesting that Are1 may play a critical role for the uptake of nitrogen on this medium. ».

5.       Section 2.2, line 134. The fusion of the gpdA promoter with Are1 was designated as the « overexpression strain » here and in numerous places of the ms (line 22, line 151, line 187, line 199), while in M &M  the authors mention « constitutive expression » based on PCR (line 337). As there is no RT-qPCR  data to confirm overexpression or to quantify the expression of this gene, the authors should delete all mentions about overexpression and constitutive expression, and they should speak only of the « gpdA ::Are1 fusion » throughout the ms.   

6.       Section 2.3 to section 2.5. All these sections use RT-qPCR ; the authors must provide MIQE compliant data [2, 3]. More specifically and according to MIQE, the authors must provide the information indicated below :

a)       Indicate why only one normalisation gene (the actin gene, see line 351) was used and provide data supporting that the expression of the actin gene is constant in all conditions used in the experiments described in this ms. Note that according to MIQE : « Normalization against a single reference gene is not acceptable unless the investigators present clear evidence for the reviewers that confirms its invariant expression under the experimental conditions described. The optimal number and choice of reference genes must be experimentally determined and the method reported. » (section 8.1 in [3]).  Please note that the usual procedure is to screen several (5 to 10) housekeeping  genes and to select among these the most stable genes in a number necessary for normalization with a dedicated software, e.g. geNorm [4]. If the authors have not checked that the actin gene is expressed at the same level in all of their experimental conditions, all their RT-qPCR experiments may be biased and most of their conclusions could be modified in future versions of the ms using correctly selected normalization genes. I guess that there is little chance that the activation effect of ARE1 on apw1 could be invalidated, as the FC is > 100 (see Figure 4A) and it hardly needs normalization genes at this level. However, most other RT-qPCR  experiments, notably those investigating the expression of glycoside hydrolase genes (Figure 6 and 7A),  are around a FC = 5 or less. These low FC require accurate normalization based on multiple reference genes [5].

b)      Indicate how many biological replicates were analyzed for each experiment.

c)       Indicate what controls were done to check RNA integrity (run RNA on gel, or Bioanalyzer, or Experion), and indicate the result of these controls.

d)      Indicate what controls were done to check that genomic contamination did not introduce a bias in cDNA quantifications (NRT control, see MIQE).

e)      Indicate what method was used for calculating the relative expression of each gene, and the reference for the method.

f)        Indicate which statistics were done to calculate p-values.

g)       Note that it is now recommended to treat p-values as continuous quantities, namely indicate the obtained value for p-value, and not if it below or above an arbitrary cut-off (see section 2 in [6]). Indicate p-values instead of subjective appreciations as « As shown in Figure 7B, there were no remarkable differences in the transcriptional levels of xyr1 and cre1 between are1 and the parental strain QM9414 » (line 227-229). Same remark for line 209-211 :  indicate p-value instead of saying that « there is no significant difference in the transcription levels of cellulase genes between Δare1 and QM9414 ».

h)      I advise the authors to add in the SI an Excel file with the Cq for each biological replicate of all conditions for each gene, plus the amplification efficiency of each gene. This may be useful for the readers, and it is a good way to keep a track of these RT-qPCR data which are easily lost upon time in the lab.

i)        Finally, I suggest to the authors to quantify the expression of Are1, Are2 and Are3 by RT-qPCR on peptone and peptone + ammonium to propose to the readers a complete view of the expression of these genes, as well as a search for Are1-binding motifs in the promoter of these genes.

7.       Section 2.4.

a)       Check whether Δare2 and Δare3 are also involved in the expression of glycoside hydrolases.

b)      Normalize results of panel B and C of Figure 5 with biomass, as Δare1 may show less FPA and EG activities because it produces less biomass than wild-type.

8.       Section 2.5. Search for Are1-binding motifs in Xyr1 and Cre1 promoters. Absence of are1-binding motifs would support that these genes are not controlled by ARE1.

9.       Section 4, M &M.

a)       Describe how was constructed the phylogenetic tree.

b)      Describe how were found the ARE1-binding motifs in the promoter regions.

c)       Describe how was performed the fermentation for 144 hours mentionned in the caption of Figure 5.  

There are numerous minor mistakes to be corrected :

1.       Line 16 : « affected » instead of « effected »

2.       Line 23 : « RT-qPCR » instead of « qRT-PCR », according to MIQE.

3.       Line 42-43 :  « Fungi have developped  efficient and sensitive… » instead of  :  « Fungi have developed the efficient and sensitive… ».

4.       Line 55 : « According to these reports » instead of « According to the light of these researches ».

5.       Line 108 : « Figure S1B and C » instead of « Figure S1B».

6.       Line 109 : « Figure S1D » instead of « Figure S1C ».

7.       Line 148 : « essential » instead of « vital ».

8.       Line 157-161 : be more specific. I suggest to replace this sentence by : « The transcription level of apw1 in T. reesei QM9414 cultured on peptone as the sole nitrogen source was increased by a fold-change of XXXX (calculate the FC) with a p-value =  XXXX (indicate p-value) when compared to apw1 level in QM9414 grown in the presence of ammonium. In the same conditions, apw2 show a fold-change of XXXX (calculate the FC) with a p-value =  XXXX (indicate p-value).

9.       Line 164-167 : compute p-values and apw1 fold-change (FC) for Δare1 on peptone and peptone + ammonium. Same for apw2 and provide values for FCs and p-values.

10.   Line 167 : « Figure 4A and B » instead of « Figure 4B ».

11.   Line 168 : « Figure 4C » instead of « Figure 3C ».

12.   Figure 5B and 5C : no error bars visible for the Δare1 strain. Is that normal ?

13.   Line 210 : « cellulase » instead of « celluase ». Check for this typo throughout the ms.

14.   Line 385 : « article » instead of « review ».

15.   Table 1 : what mutant was constructed with primers GFP-1-F1 and GFP-717-R1 ?

16.   References : many species names should be in italics (e. g. Trichoderma reesei in line 401, and 411; Candida albicans  in line 455 and 456 ; Microsporum canis in line 474 ).

17.   Supplementary Figures : too many mistakes to be corrected by a reviewer !!!! Ladders are badly adjusted, some figures are inverted, what is TrRas1 making in the caption of Figure S1, part of caption of Figure S1 is present in caption of Figure S2 and S3 and confusing the reader, incorrect naming of primers,…. Look at the figures, read the captions carefully, and make corrections !! Indicate expected sizes for PCR fragments and reference for commercial ladder.

References

1.            Grossetete, S., B. Labedan and O. Lespinet, FUNGIpath: a tool to assess fungal metabolic pathways predicted by orthology. BMC Genomics, 2010. 11: p. 81.

2.            Bustin, S.A., Why the need for qPCR publication guidelines?--The case for MIQE. Methods, 2010. 50(4): p. 217-26.

3.            Bustin, S.A., V. Benes, J.A. Garson, J. Hellemans, J. Huggett, M. Kubista, R. Mueller, T. Nolan, M.W. Pfaffl, G.L. Shipley, J. Vandesompele and C.T. Wittwer, The MIQE guidelines: minimum information for publication of quantitative real-time PCR experiments. Clin Chem, 2009. 55(4): p. 611-622.

4.            Vandesompele, J., K. De Preter, F. Pattyn, B. Poppe, N. Van Roy, A. De Paepe and F. Speleman, Accurate normalization of real-time quantitative RT-PCR data by geometric averaging of multiple internal control genes. Genome Biol, 2002. 3(7): p. RESEARCH0034.

5.            Vandesompele, J., M. Kubista and M.W. Pfaffl, Reference gene validation software for improved normalization, in Real-time PCR current technology and applications, J. Logan, K. Edwards, and N. Saunders, Editors. 2009, Caister Academic Press: Norfolk UK. p. 47-64.

6.            Wasserstein, R.L., A.L. Schirm and N.A. Lazar, Moving to a World Beyond “p < 0.05”. The American Statistician, 2019. 73: sup1: p. 1-19.
